# Prognostic Factors in Epithelioid Hemangioendothelioma: Analysis of a Nationwide Molecularly/Immunohistochemically Confirmed Cohort of 57 Cases

**DOI:** 10.3390/cancers15133304

**Published:** 2023-06-23

**Authors:** Tess Tomassen, Yvonne M. H. Versleijen-Jonkers, Melissa H. S. Hillebrandt-Roeffen, Patricia H. J. Van Cleef, Thijs van Dalen, Marije E. Weidema, Ingrid M. E. Desar, Uta Flucke, Joost M. van Gorp

**Affiliations:** 1Department of Pathology, Radboud University Medical Center, Geert Grooteplein Zuid 10, 6525 GA Nijmegen, The Netherlands; patricia.vancleef@radboudumc.nl (P.H.J.V.C.); uta.flucke@radboudumc.nl (U.F.); 2Department of Medical Oncology, Radboud University Medical Center, Geert Grooteplein Zuid 10, 6525 GA Nijmegen, The Netherlands; yvonne.versleijen-jonkers@radboudumc.nl (Y.M.H.V.-J.); melissa.hillebrandt-roeffen@radboudumc.nl (M.H.S.H.-R.); marije.weidema@radboudumc.nl (M.E.W.); ingrid.desar@radboudumc.nl (I.M.E.D.); 3Department of Surgery, Utrecht University Medical Center, Heidelberglaan 100, 3584 CX Utrecht, The Netherlands; t.dalen-4@umcutrecht.nl; 4Princess Máxima Center for Pediatric Oncology, Heidelberglaan 25, 3584 CS Utrecht, The Netherlands; 5Department of Pathology, St. Antonius Hospital, Koekoekslaan 1, 3435 CM Nieuwegein, The Netherlands; j.van.gorp@antoniusziekenhuis.nl

**Keywords:** epithelioid hemangioendothelioma, aggressive, malignant, survival, prognosis, risk stratification, lymph node

## Abstract

**Simple Summary:**

Epithelioid hemangioendothelioma (EHE) is a very rare malignant vascular neoplasm with unpredictable clinical course due to its remarkable heterogeneity. Large cohort studies with molecularly/immunohistochemically confirmed EHE are scarce, and what determines survival has been controversial. This retrospective nationwide cohort study addresses EHE epidemiology and aims to identify clinical and histopathological parameters with prognostic significance, thereby providing useful insights into the clinical behavior of this rare cancer. Our findings emphasize the aggressive behavior of EHE, demonstrated by lower 1- and 5-year overall survival rates compared to those in the current literature. Moreover, we confirmed the usefulness of the risk stratification model by Shibayama for unifocal disease and showed that multifocal and metastatic disease have no survival differences, indicating that multifocality is early metastatic disease. Tumors in the head and neck area have a higher propensity for lymph node metastases, entailing consideration of lymphadenectomy.

**Abstract:**

Epithelioid hemangioendothelioma (EHE) is an extremely rare vascular sarcoma with variable aggressive clinical behavior. In this retrospective study, we aimed to investigate prognostic factors based on clinicopathologic findings in a molecularly/immunohistochemically confirmed nationwide multicenter cohort of 57 EHE cases. Patients had unifocal disease (*n* = 29), multifocal disease (*n* = 5), lymph node metastasis (*n* = 8) and/or distant metastasis (*n* = 15) at the time of diagnosis. The overall survival rate was 71.4% at 1 year and 50.7% at 5 years. Survival did not correlate with sex, age or histopathological parameters. No survival differences were observed between multifocal and metastatic disease, suggesting that multifocality represents early metastases and treatment options are limited in comparison to unifocal disease. In unifocal tumors, survival could be predicted using the risk stratification model of Shibayama et al., dividing the cases into low- (*n* = 4), intermediate- (*n* = 15) and high- (*n* = 3) risk groups. No clinical or histopathological parameters were associated with progressive unifocal disease course. Lymph node metastases at the time of diagnosis occurred in 14.0% of the cases and were mainly associated with tumor localization in the head and neck area, proposing lymph node dissection. In conclusion, our results demonstrate the aggressive behavior of EHE, emphasize the prognostic value of a previously described risk stratification model and may provide new insights regarding tumor focality, therapeutic strategies and prognosis.

## 1. Introduction

Epithelioid hemangioendothelioma (EHE) is an extremely rare vascular sarcoma, with an estimated prevalence of less than one in a million people [1]. It originates from precursor cells with endothelial properties and specific fusion genes [2,3,4,5]. EHE has a peak incidence in middle age; however, the range is broad, with children also being affected [1,6,7]. The tumor can occur in any part of the body but commonly affects the lungs, liver and soft tissue and more rarely bone [1,5,8,9]. 

Diagnosis is made based on histologic, immunohistochemical and molecular characteristics. Histological features commonly consist of relatively monomorphic epithelioid cells arranged in cords and nests within a myxohyaline stroma. The cells typically show very subtle intracytoplasmic lumina presenting as vacuoles [5,9,10]. Cellular atypia with pleomorphism is rarely seen and mitotic figures are sparse [5]. Positive staining for endothelial markers such as ERG and CD31 and specific staining for CAMTA1 or TFE3 identify EHE immunohistochemically [8,11,12,13]. The genetic hallmark is a WWTR1-CAMTA1 or YAP-TFE3 gene fusion, detected in nearly 90% and 10% of the cases, respectively [8,14,15,16]. 

The clinical behavior of EHE was initially considered as intermediate/borderline between hemangioma and conventional angiosarcoma [9]. Later, the World Health Organization (WHO) reclassified EHE as fully malignant, thereby acknowledging its aggressive nature [5]. The tumor rarity, broad age range of patients, variable clinical presentation with different anatomical sites and multifocality in visceral organs make it hard to define consolidated risk factors. Clinical and pathological parameters associated with survival have only been described in a handful of large studies [1,17,18]. Other studies performed statistical analyses on smaller subgroups, reducing the reliability of the statistical outcomes [10,19,20]. 

In this retrospective study of a molecularly/immunohistochemically confirmed cohort of 57 EHE cases collected from across the country, prognostic factors based on clinicopathologic characteristics were determined.

## 2. Materials and Methods

### 2.1. Tissue Collection and Clinical Data

Through searching PALGA (Dutch nationwide network and registry of histo- and cytopathology), data and formalin-fixed paraffin-embedded (FFPE) material from available EHE cases were collected. In order to receive clinical information (sex, age at diagnosis, tumor site and size, treatment details and follow-up) data were anonymously extracted from the Netherlands Cancer Registry (NCR). Missing clinical information was requested anonymously from the participating hospitals. Additionally, more recent cases from the Radboud University Medical Centre were included. Ethical approval for this study was obtained from the local certified Medical Ethics Committee of the Radboud University Medical Center, Nijmegen, The Netherlands (file number: 2018-4610). Fifteen cases were previously published [4,21]. 

Cases were histologically reviewed. Additional immunohistochemistry and/or molecular tests (see below) were performed to confirm the diagnosis. 

Tumor histology was classified as typical and atypical, in keeping with the definition by Shibayama et al. and Rosenbaum et al. [17,18]. Atypical histology was determined by at least two of the following criteria: high mitotic activity (>1/2 mm^2^), high nuclear grade and tumor necrosis. High nuclear grade was defined as the presence of enlarged, round and swollen nuclei with vesicular chromatin and prominent nucleoli. Manual mitotic counting was performed with a standard area of 2.37 mm^2^ (40× objective and 10× ocular with field number 22 mm). The number of mitoses was scored based on 10 high-power fields (HPFs). Cases with fewer than 10 HPFs were excluded from counting.

Tumor size was determined based on macroscopy, histological slides and/or radiology. Multifocality was defined as multiple tumor nodules limited to one (visceral) organ at the time of diagnosis, whereas metastatic disease, syn- or metachronous, included (regional) lymph node involvement and deposits in visceral organs and all other sites.

The follow-up end point was the time between initial diagnosis and the date of death or the time between diagnosis and the last follow-up. 

### 2.2. Molecular Analysis 

#### 2.2.1. Fluorescence In Situ Hybridization (FISH)

Sections of 4 µm FFPE tissue sections were submitted to FISH analysis. For the FISH process, 10 µL SPEC TFE3 probe (ZytoLight® SPEC TFE3 Dual Color Break Apart Probe, z-2109, Zytovision, Bremerhaven, Germany) and CAMTA1 probe (CAMTA1 Split FISH probe, FS0035, Abnova, Taipei, Taiwan) were applied to the pre-treated slides. Finally, the slides were mounted with a solution containing both DAPI and Vectashield (Vector, Brunschwig, Amsterdam, The Netherlands). TFE3 and CAMTA1 signals were scored using a Leica DMRBE (Leitz, Grand Rapids, MI, USA) fluorescence microscope. At least 50 nuclei per sample were counted and were scored as negative (<20%) or positive (≥20%). 

#### 2.2.2. Reverse Transcriptase–Polymerase Chain Reaction (RT-PCR)

Extracted RNA was submitted to RT-PCR analysis as described previously by Flucke et al. (2014). For detection of the t(1;3)(p26.3;q25) translocation, a WWTR1 exon3 and exon4 forward primer and a CAMTA1 exon8 reverse primer were used. For detection of YAP1-TFE3, the YAP1 exon1 forward primer and TFE3 exon4, exon6, exon8 and exon10 reverse primer were used [4]. 

#### 2.2.3. Immunohistochemistry

CAMTA1 (NBP1—93620, Novusbio, Centennial, CO, USA) and TFE3 (HPA023881, Sigma-Aldrich, St. Louis, MO, USA) immunohistochemistry staining was performed on 4 µm FFPE tissue sections when both RT-PCR and FISH could not be performed or were not interpretable.

### 2.3. Statistical Analysis 

SPSS software 25.0 was used for data analyses. The Kaplan–Meier estimate was used to determine the overall cumulative survival. The statistical significance of different variables (sex, age at diagnosis, tumor size, histology, focality) in relation to survival and progression was determined using log-rank analyses. The Kruskal–Wallis test was used to compare groups. Statistical analyses were considered as significant for any value of *p* less than 0.05.

## 3. Results

### 3.1. Patient Demographics

Patient characteristics are described in Table 1. The tumors occurred in adults (93.0%, *n* = 53) and children (7.0%, *n* = 4) of both sexes, with a slight female predominance (female 52.6%, *n* = 30; male 47.4%, *n* = 27). Age at diagnosis ranged from 9 to 87 years, with a median of 54 years.

### 3.2. Tumor Features 

The majority of the cases had unifocal disease at time of diagnosis (50.9%, *n* = 29). Multifocal and metastatic disease at time of diagnosis were seen in 5 (8.8%) and 23 (40.4%) cases, respectively. Metastatic disease included both lymph node (14.0%, *n* = 8) and multi-organ (26.3%, *n* = 15) involvement. 

Unifocal soft tissue lesions (62.1%, *n* = 18) arose in the lower extremities (20.7%, *n* = 6), thorax (17.2%, *n* = 5), upper extremities (10.3%, *n* = 3), groin (6.9%, *n* = 2) and soft tissue of the head and neck area (6.9%, *n* = 2). Other unifocal localization was in the long bones (of the upper arm and lower leg) (13.8%, *n* = 4), lungs (10.3%, *n* = 3), liver (3.4%, *n* = 1), skin (of the thumb) (3.4%, *n* = 1), ear (3.4%, *n* = 1) and lymph node (3.4%, *n* = 1) (Figure 1). The median tumor size in unifocal disease was 3.5 cm (range 1.1–11.0, *n* = 19). For seven cases, tumor size was only defined as <3 cm (*n* = 2), >3 cm (*n* = 2), <5 cm (*n* = 2) or >5 cm (*n* = 1). 

Multifocal disease originated in the liver (80.0%, *n* = 4) and lungs (20.0%, *n* = 1). Within the lymph node metastases group, primary localization comprised the clavicular region (37.5%, soft tissue *n* = 2, bone *n* = 1), mandibular bone (25.0%, *n* = 2), liver (12.5%, *n* = 1), pelvic soft tissue (12.5%, *n* = 1) and pleura (12.5%, *n* = 1). 

The exact localizations of distant metastases are shown in Figure 1, with two cases (case 47 and 48) being unknown.

### 3.3. Treatment

For unifocal disease, all but three cases were treated by surgery (86.2%, *n* = 25). Additional radiotherapy was given in six (20.7%) cases (adjuvant *n* = 4, neo-adjuvant *n* = 1, unknown *n* = 1). The remaining patients received radiotherapy only (due to irresectable tumor or because of the patient’s age) or followed an expectative policy. For one patient with unifocal disease, primary treatment details were lacking. 

Patients with multifocal localization underwent hemihepatectomy (40.0%, *n* = 2), systemic therapy (40.0%, *n* = 2) or had a wait and see approach (20.0%, *n* = 1). All but one patient with lymph node metastases underwent surgery (87.5%, *n* = 7), either as monotherapy (37.5%, *n* = 3), with radiation (37.5%, adjuvant *n* = 2, unknown *n* = 1) or with systemic therapy (12.5%, *n* = 1). Other treatment within the lymph node metastatic group comprised radiotherapy only (due to irresectable tumor, 12.5%, *n* = 1). 

For distant metastases, treatment with systemic therapy was most common (66.7%, *n* = 10) including three patients receiving additional surgical intervention (20.0%), followed by radiotherapy (20.0%, *n* = 3), surgery only (6.7%, *n* = 1) and expectative policy (6.7%, *n* = 1). Treatment details about systemic therapy are listed in Appendix A. 

### 3.4. Histopathological Characteristics

Histological characteristics (*n* = 53) including nuclear grade, mitotic count and necrosis are shown in Table 2. Necrosis and high nuclear grade were seen in 18.9% (*n* = 10) and 37.7% (*n* = 20) of the cases, respectively. Mitotic count >1/10 HPF was found in 37.7% (*n* = 20) of the tumors. Atypical tumor histology, based on the classification of Shibayama et al. and Rosenbaum et al., was observed in 30.2% (*n* = 16) of the cases [17,18]. 

Of all cases, 24 (42.1%) were confirmed using RT-PCR, resulting in 23 WWTR1-CAMTA1 and 1 YAP-TFE3 fusion-positive cases. A total of 21 cases showed rearrangements for CAMTA1 (36.8%) and 1 for TFE3 (1.8%) using FISH. The remaining 11 cases (19.3%) were positive for CAMTA1 immunohistochemistry (Table 2). 

### 3.5. Survival 

Follow-up was available for all cases. After a median follow-up time of 35 months (range 0–324 months), 27 patients were alive (no evidence of disease (NED) 35.1%, *n* = 20; alive with disease (AWD) 12.3%, *n* = 7) and 30 patients had died (dead of disease (DOD) 50.9%, *n* = 29; dead due to other cause (DOC) 1.8%, *n* = 1). 

The median follow up time in the survivor group was 70.0 months (range 13–324 months), and patients died after a median period of 11.5 months (range 0–282 months). The overall 1- and 5-year survival rate was 71.4% and 50.7%, respectively. The Kaplan–Meier plot shows the overall survival for 56 patients (Figure 2). One patient died as result of endometrial cancer and was therefore excluded from survival analyses. 

Survival was unaffected by sex (*p* = 0.839), age at diagnosis (<55 or ≥55, *p* = 0.897), necrosis (*p* = 0.097), nuclear grade (*p* = 0.759), mitosis (≤1/10 HPF or >1/10 HPF, *p* = 0.298) and atypical histology (*p* = 0.198). Unifocal disease showed better survival compared to lymph node (*p* = 0.013) and multi-organ (*p* ≤ 0.001) metastatic disease. Patients with multifocal disease had the same prognosis compared to unifocal disease (*p* = 0.547), lymph node metastasis (*p* = 0.561) and multi-organ metastasis (*p* = 0.087). Overall survival (*p* = 0.595) and disease free survival (*p* = 0.324) did not differ between the different unifocal localizations. Concerning the multi-organ metastatic group, patients with pleura involvement had significantly worse outcome (*p* = 0.012). Neither lung (*p* = 0.062) or bone involvement (*p* = 0.839) was correlated with outcome.

### 3.6. Progressive Unifocal Disease 

Among the unifocal EHEs, progressive clinical course was observed in 10 cases (41.7%), with progression of primary tumor (8.3%, *n* = 2), residual disease (8.3%, *n* = 2), metastatic disease (12.5%, *n* = 3) and residual and metastatic disease (12.5%, *n* = 3). Fourteen patients (58.3%) had no progressive disease. For five patients, data about progression were lacking. The majority of the unifocal progression group was treated with surgery (either with or without radiotherapy, 90.0% *n* = 9). In five of them (case 23, 36, 41, 44, 49), surgical margins were insufficient (defined as either positive margin or margin <5 mm). 

Statistical analyses were performed to determine clinical or histopathological parameters related to progressive unifocal disease course. Tumor size ≥3.0 cm (*p* = 0.544), mitotic number ≥1/10 HPF (*p* = 0.937), the presence of necrosis (*p* = 0.133), nuclear grade (*p* = 0.726) and atypical histology (*p* = 0.137) were not associated with a progressive clinical course. 

### 3.7. Risk Stratification Model of Shibayama et al.

To predict overall survival, patients with unifocal EHE localization were assigned based on tumor size (≤3 cm vs. >3 cm) and histology (atypical vs. typical), stratifying 22 tumors into low-risk (18.2%, *n* = 4), intermediate-risk (68.2%, *n* = 15) and high-risk (13.6%, *n* = 3) groups, according to the classification of Shibayama et al. [18]. The other unifocal cases could not be implemented into the scoring system due to missing data (*n* = 6) or because of another cause of death (*n* = 1) (Table 3). All patients assigned to the low-risk group were alive during follow-up (NED, *n* = 3; AWD, *n* = 1). The clinical course of the intermediate-risk category was variable (NED, *n* = 9; AWD, *n* = 1; DOD, *n* = 5) and no disease-free patients were present in the high-risk group (AWD, *n* = 1; DOD, *n* = 2). 

Survival differences were significant between the three risk groups (*p* = 0.028). However, tumor size (*p* = 0.065) and atypical histology (*p* = 0.254) as independent parameters were not associated with worse prognosis.

## 4. Discussion

In 1982, Weiss and Enzinger coined the term EHE to describe a low-grade malignant vascular tumor now considered fully malignant [5,9]. Several reports reflect the remarkable heterogeneity of EHE. Therefore, identifying independent prognostic factors is challenging and often contradictory among studies [1,10,17,18,19,22]. In the present investigation, we searched for prognostically relevant clinical and histopathological parameters in a molecularly or immunohistochemically confirmed cohort of 57 EHE patients. 

Large datasets with regard to optimal treatment strategy are limited. However, if possible, surgical resection with wide margins should be performed [8,23]. More than half of our unifocal cases lacking sufficient surgical margins consequently showed progressive disease course. Based on three cases, we also confirmed the importance of surgical interventions for the management of multifocal hepatic EHE [24].

Tumor size and mitotic count appeared to be important predictive values and were both included in the risk stratification models of Shibayama et al. and Deyrup et al. [10,18]. However, the model of Shibayama et al. includes the size of multifocal lesions, assuming that a multifocal lesion equals unifocal disease. One could argue that multifocal EHE is early metastatic disease rather than simultaneous independent origin of multiple lesions [25,26]. This is supported by our finding that there is no significant difference in survival between the groups of multifocal and metastatic disease. Nevertheless, we could not statistically distinguish unifocal from multifocal disease. It might be that primary treatment is a confounder influencing survival, thereby affecting the statistical difference between both groups (uni- versus multifocal). Also, the multifocal cohort needs to be larger to perform proper (statistical) analysis because of the heterogeneity. 

However, when assuming that multifocal EHE represents early metastases, it seems reasonable to enter only unifocal tumor size into a prognostic model. We replicated the risk stratification from Shibayama et al. on the unifocal EHE group and confirmed the prognostic value of tumor size and atypical histology [18]. Both parameters were not independently associated with shorter survival, highlighting the limitation of our findings. Also, when creating a reliable diagnostic model, ideally, the whole tumor should be histologically assessed instead of limited tissue (mostly provided by external laboratories, at least in our study).

Our population showed a lower 1- and 5-year overall survival rate (71.4% and 50.7%, respectively) compared to most previous studies, which report 1- and 5-year overall survival rates between 89.0 and 96.2% and 68.0 and 78.8%, respectively [1,10,18,20,27]. We are aware of the fact that EHE populations are always very heterogeneous and therefore difficult to compare. Unlike other publications, Rosenbaum et al. demonstrated survival rates similar to ours, based on a molecularly confirmed EHE population including 93 cases [17]. In both ours and Rosenbaum’s study population, metastatic disease (syn- or metachronous) was seen in half of the cases, whereas EHE was initially thought to metastasize in only 20–30% of the cases [9,10,17]. This higher mortality number might be affected by several reasons. First, our cohort is too small when heterogeneity with variable behavior is taken into account. Also, it might be possible that previous, especially older studies, accidently excluded atypical morphological cases mimicking angiosarcoma or other neoplasms when specific immunohistochemical and molecular analyses were lacking, whereas we confirmed the diagnoses in all cases molecularly/immunohistochemically, which is ed possibly reflected in lower overall survival rates. Furthermore, the majority of the cases were treated at an academic center, likely causing some bias with more complex cases including unifocal tumors without complete resection and with progressive disease course. 

Due to the heterogeneity of the study population of Rosenbaum et al. and ours, no overlying factors affecting survival could be observed. Within our cohort, besides the high metastatic number of cases, there is no clear explanation or causal relation between different parameters and survival. Previously described clinical parameters related to worse outcome such as pulmonary and/or pleura involvement did not predominate within this cohort, suggesting that the aggressiveness is likely due to the tumor biology. It might be that secondary (epi)genetic alterations enhance tumor progression, demonstrated by a lower survival rate within our study [28,29]. 

We classified cases with lymph node metastases as a separate group. In the literature, lymph node metastatic disease is rarely described in larger subsets or as a solitary category [17]. Despite our small number of cases with lymph node metastases, it is striking that lymph node involvement mainly occurred in tumors originating in the head and neck region. This is likely due to the close relationship of anatomic structures including naturally lymphatic vessels. Since lymph node metastases show significantly poorer survival compared to unifocal disease, preventing lymph node spread through locoregional lymph node dissection should be recommended, especially in the head and neck area [17,30]. When looking at the group with distant metastases at the time of diagnosis, all cases showed either lung or liver involvement. It is hypothesized that these organs provide fertile ground for metastatic disease because of their good vascularization [31].

## 5. Conclusions

In conclusion, we performed a nationwide multi-center study in a heterogenous population of 57 EHE cases. We showed a 1- and 5-year overall survival rate that is approximately 20% lower than the average based on previous articles, demonstrating the aggressive behavior of a subset of EHE. The risk stratification by Shibayama et al. is a good predictor for unifocal disease [18]. As expected, multifocality and metastatic behavior have a similarly bad prognosis. Tumors of the head and neck have a relatively high propensity for lymph node metastases, which should have clinical implications. Given the rarity of EHE, this study enriches the scarce literature.

## Figures and Tables

**Figure 1 cancers-15-03304-f001:**
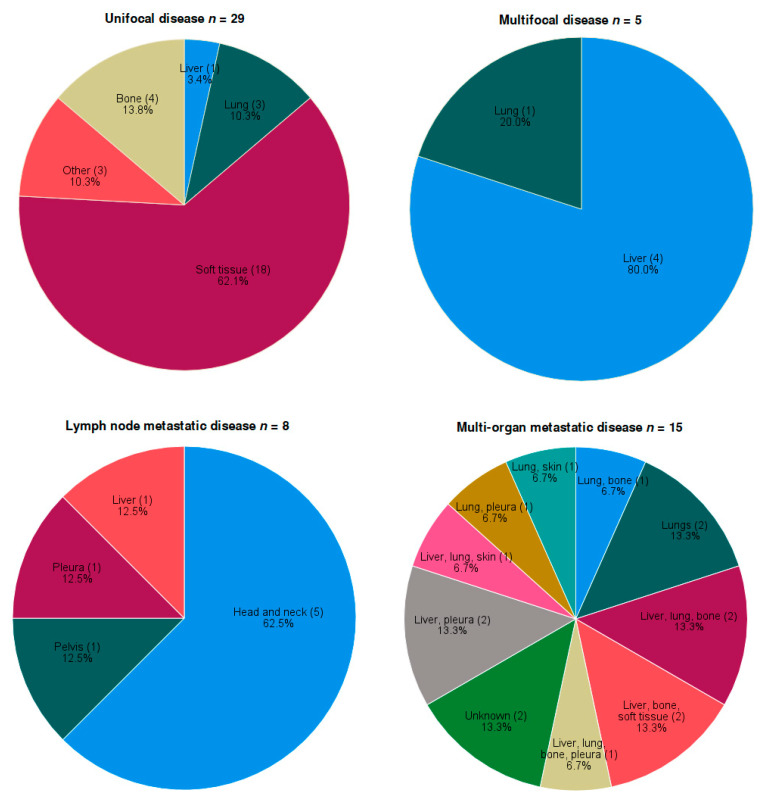
Localization of epithelioid hemangioendothelioma at time of diagnosis.

**Figure 2 cancers-15-03304-f002:**
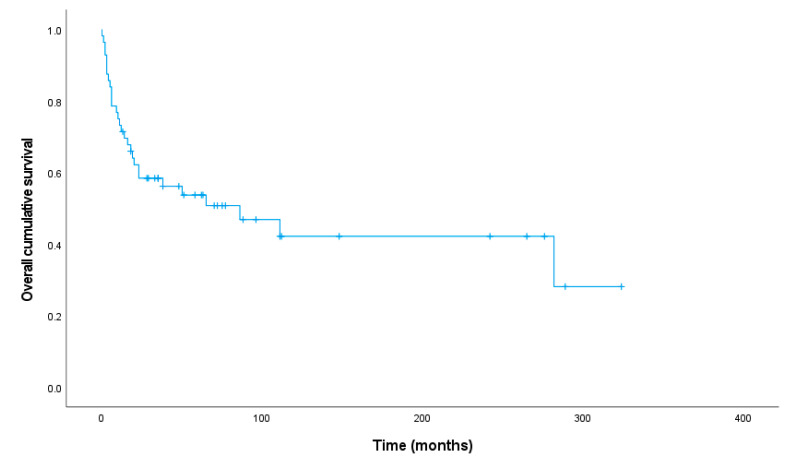
Disease-specific overall survival for patients with epithelioid hemangioendothelioma.

**Table 1 cancers-15-03304-t001:** Baseline characteristics of the entire epithelioid hemangioendothelioma cohort.

Case	Sex/Age (Years)	Site of Primary Tumor (Focality)	Size (Centimeters)	Therapy	R/Met/P (Months)	Current Life Status, Follow-Up (Months)
1	F/9	Ear ** (uf)	<3.0	E	R, 25	NED, 276
2	F/14	Upper arm, soft tissue (uf)	<3.0	E	No	NED, 35
3	M/20	Supraclavicular, soft tissue (met-L)	2.4	E + RT (adj)	No	NED, 38
4	F/26	Upper arm, bone (uf)	NA	E	No	NED, 62
5	M/34	Upper leg, soft tissue (uf)	6.0	E	No	NED, 242
6	F/37	Skin of the thumb (uf)	NA	E	No	NED, 72
7	F/37	Liver (mf)	2.5	E	No	NED, 63
8	M/40	Upper leg, soft tissue (uf)	<5.0	E + RT (adj)	No	NED, 324
9	F/41	Groin, soft tissue (uf)	1.8	E + RT (adj)	Met, 35	NED, 35
10	F/49	Mandibula, bone (met-L)	2.5	E	No	NED, 88
11	M/59	Heart (uf)	8.5	E	No	NED, 96
12	F/59	Lymph node ** (uf)	2.0	E + RT (adj)	Met, 57	NED, 148
13	M/59	Axilla, soft tissue (uf)	4.5	E + RT (neo-adj)	No	NED, 70
14	M/59	Groin, soft tissue (uf)	1.1	E	No	NED, 51
15	F/63	Parotic gland (uf)	1.6	E	No	NED, 29
16	F/66	Liver (mf)	NA	E	No	NED, 75
17	F/67	Upper leg, soft tissue (uf)	11.0	E	No	NED, 33
18	M/69	Upper arm, soft tissue (uf)	1.5	E	No	NED, 289
19	F/69	Liver (uf)	3.5	E	NA	NED, 48
20	M/71	Mediastinum (uf)	10.0	E	No	NED, 58
21	M/19	Lung (met-M) *	NA	S	No	AWD, 112
22	F/33	Skin of the scalp (met-M)	2.0	Ex	No	AWD, 13
23	F/41	Supraclavicular, soft tissue (uf)	2.5	E + RT (adj)	Met, 19, 44, 68	AWD, 111
24	M/54	Lung (uf)	2.2	E	NA	AWD, 265
25	M/63	Lower leg, bone (uf)	3.4	E	R + Met, 13	AWD, 77
26	M/63	Liver (mf)	NA	Ex	P, 10	AWD, 18
27	M/66	Lung (met-M)	NA	E	NA	AWD, 28
28	F/9	Lung (mf)	NA	S	P, 3	DOD, 6
29	F/10	Lung (met-M)	11.0	E + S	P, 4	DOD, 6
30	F/32	Skin of the scalp (met-M)	3.0	E + S	NA	DOD, 11
31	M/33	Pleurae (met-L)	NA	E + S	Met, 3	DOD, 6
32	M/37	Lung (uf)	>3.0	RT	NA	DOD, 16
33	M/39	Supraclavicular, soft tissue (met-L)	NA	RT	P, 8	DOD, 12
34	M/39	Lung (met-M)	5.2	RT	No	DOD, 1
35	M/39	Mediastinum (met-M)	5.3	S	P + Met, 7	DOD, 9
36	M/42	Upper leg, soft tissue (uf)	10.0	E	R + Met, 17	DOD, 20
37	M/42	Lung, liver, bone (met-M) *	NA	S	Met, 19	DOD, 23
38	M/45	Lung (met-M)	NA	S	NA	DOD, 3
39	M/49	Mandibula, bone (met-L)	3.1	E + RT (adj)	R + Met, 22	DOD, 111
40	F/50	Pleurae (met-M)	NA	S	Met, 4	DOD, 5
41	M/52	Thorax, soft tissue (uf)	NA	E + RT (unknown)	P, 19	DOD, 19
42	F/53	Liver (mf)	NA	S	Met, 2	DOD, 86
43	F/54	Liver (met-L)	NA	E	NA	DOD, 14
44	F/55	Upper leg, bone (uf)	7.5	E	Met, 9	DOD, 10
45	F/55	Liver and pleurae (met-M) *	NA	S	NA	DOD, 3
46	M/62	Clavicular, bone (met-L)	2.2	E + RT (unknown)	Met, 7	DOD, 18
47	F/67	Lower leg, soft tissue (met-M)	>5.0	E + S	NA	DOD, 23
48	M/67	Sternum (met-M)	>5.0	RT	NA	DOD, 3
49	M/68	Lung (uf)	1.4	E	R, 13	DOD, 50
50	F/68	Pelvis, soft tissue (met-L)	NA	E	Met, 35	DOD, 38
51	F/70	Pleurae (met-M)	NA	S	NA	DOD, 4
52	F/81	Upper leg, soft tissue (uf)	9.0	NA	NA	DOD, 0
53	M/82	Mediastinum (uf)	>3	Ex	No	DOD, 2
54	F/82	Shoulder, soft tissue (met-M)	11.0	RT	NA	DOD, 2
55	F/56	Mediastinum (uf)	>5.0	E	NA	DOD, 282
56	F/87	Upper leg, soft tissue (uf)	<5.0	RT	P, 9	DOD, 65
57	F/55	Rib (uf)	4.5	E	No	DOC, 139

uf, unifocal disease; mf, multifocal disease; R, recurrence; P, progression primary lesion; Met, metastasis; Met-L, lymph node metastasis; Met-M, multi-organ metastasis; NA, not available; E, excision; Ex, expectative; RT, radiotherapy; S, systemic therapy; NED, no evidence of disease; AWD, alive with disease; DOD, dead of disease; DOC, dead due to other cause; Adj, adjuvant; Neo-adj, neo-adjuvant. * Primary localization unknown. ** Not further specified.

**Table 2 cancers-15-03304-t002:** Histological, immunohistochemical and molecular data.

Case	Necrosis	Nuclear Grade	Mitosis/10 HPF	RT-PCR	FISH	Immunohistochemistry
1	No	Low	0	-	-	CAMTA
2	Yes	High	1	Neg	-	CAMTA
3	No	High	0	*WWTR1-CAMTA1*	-	-
4	No	Low	0	*WWTR1-CAMTA1*	*-*	-
5	No	Low	1	-	*CAMTA*	-
6	No	High	0	-	*CAMTA*	-
7	No	Low	1	*WWTR1-CAMTA1*	*-*	-
8	No	Low	4	-	Neg	CAMTA
9	Yes	High	0	-	*CAMTA*	-
10	No	High	10	-	*CAMTA*	-
11	No	Low	2	-	*TFE3*	
12	No	Low	1	-	*CAMTA*	-
13	No	High	0	-	*CAMTA*	-
14	NA	NA	NA	*WWTR1-CAMTA1*	-	-
15	No	High	2	*WWTR1-CAMTA1*	-	-
16	No	High	0	-	*CAMTA*	-
17	No	Low	3	*WWTR1-CAMTA1*	-	-
18	No	Low	0	-	*CAMTA*	-
19	No	Low	2	-	*CAMTA*	-
20	No	Low	2	*WWTR1-CAMTA1*	-	-
21	No	High	0	*YAP1-TFE3*	-	-
22	No	Low	0	*WWTR1-CAMTA1*	-	-
23	No	Low	1	*WWTR1-CAMTA1*	-	-
24	Yes	High	0	*WWTR1-CAMTA1*	*-*	-
25	No	High	8	*WWTR1-CAMTA1*	-	-
26	NA	NA	NA	*WWTR1-CAMTA1*	-	-
27	No	Low	0	-	*CAMTA*	-
28	No	Low	0	*WWTR1-CAMTA1*	*-*	-
29	Yes	Low	6	-	*CAMTA*	-
30	Yes	High	18	-	NI	CAMTA
31	Yes	High	2	*WWTR1-CAMTA1*	-	-
32	No	Low	0	-	*CAMTA*	-
33	No	Low	1	-	-	CAMTA
34	No	High	2	*WWTR1-CAMTA1*	*-*	-
35	No	Low	1	*WWTR1-CAMTA1*	-	-
36	No	Low	7	-	*CAMTA*	-
37	No	Low	1	-	*CAMTA*	-
38	No	High	0	-	*CAMTA*	-
39	No	Low	0	*WWTR1-CAMTA1*	*-*	-
40	No	Low	0	*WWTR1-CAMTA1*	-	-
41	NA	NA	NA	-	-	CAMTA
42	No	High	4	-	-	CAMTA
43	No	Low	2	-	-	CAMTA
44	No	Low	4	*WWTR1-CAMTA1*	NI	-
45	No	Low	0	Neg	*CAMTA*	-
46	No	High	2	-	*CAMTA*	-
47	Yes	Low	3	-	-	CAMTA
48	No	Low	6	-	*CAMTA*	-
49	Yes	High	0	*WWTR1-CAMTA1*	*-*	-
50	NA	NA	NA	-	-	CAMTA
51	No	Low	0	*WWTR1-CAMTA1*	-	-
52	Yes	High	1	*WWTR1-CAMTA1*	*-*	-
53	No	High	3	Neg	*CAMTA*	-
54	Yes	Low	1	-	-	CAMTA
55	No	Low	0	-	*CAMTA*	-
56	No	Low	0	-	*CAMTA*	-
57	No	Low	0	*WWTR1-CAMTA1*	*-*	-

NA, not available; NI, not interpretable; Neg, negative.

**Table 3 cancers-15-03304-t003:** Implementation of the risk stratification model of Shibayama et al. in cases with unifocal disease, stratifying the risk into low (total score 0), intermediate (total score 1) or high (total score 2).

Case *	Tumor Size > 3 cm	Atypical Histology	Score
1	No	No	0
2	No	Yes	1
4	-	No	-
5	Yes	No	1
6	-	No	-
8	-	No	-
9	No	Yes	1
11	Yes	No	1
12	No	No	0
13	Yes	No	1
14	No	-	-
15	Yes	No	1
17	No	Yes	1
18	Yes	No	1
19	No	No	0
20	Yes	No	1
23	No	No	0
24	No	Yes	1
25	Yes	Yes	2
32	Yes	No	1
36	Yes	No	1
41	-	-	-
44	Yes	No	1
49	No	Yes	1
52	Yes	Yes	2
53	Yes	Yes	2
55	Yes	No	1
56	-	No	-

* Case 57 is not included due to another cause of death.

## Data Availability

The authors will provide the data upon request.

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
