# Peer review of "Prognostic Factors in Epithelioid Hemangioendothelioma: Analysis of a Nationwide Molecularly/Immunohistochemically Confirmed Cohort of 57 Cases"

_cancers, 2023, doi:10.3390/cancers15133304_

Round 1

Reviewer 1 Report

Authors describe EHE prognostic factors , while the study does include larger cohort it is marred by hetergeneity in a retrospective study. Some comments

1)While IHC for CAMTA1 is established method, a comparision of WWTR1-CAMTA1 with FISH would have provided useful data but clearly only cases that were positive for any one were taken. 

One image of atypical vs typical histology may provide readers with more information 
2) EHE from different sites have different prognosis like for e.g LIver EHE have indolent behavior, breast EHE is also better behvaiour Vs Lung EHE. This is not evident in the above study however the details in the table indicate that non lung EHE have better Relapse free survival. Can a Kaplan meier analysis of Site wise comparision for RFS or DFS be done. 

Also size appears to be a factor as all non lung tumors who died have size > 5 cm or size not available. Is recalculation for same possible. Amongst patients who are alive non lung tumors dominate.. Also details on chemotherapy in metastatic patients if available should be given. 

Reviewer 2 Report

Dear Editor,

I read this manuscript with great interest. Although, Epithelioid hemangioendothelioma (EHE) is an extremely rare vascular sarcoma, the size of the study population is sufficient and findings are clearly presented. 

To my point of view,  this manucript has important results and can be published in this journal.

Author Response

We thank the reviewer for the very generous respond on our article. There were no specific suggestions, therefore no specific feedback has been processed. 

Reviewer 3 Report

Tomassen et al. investigated prognostic factors of epithelioid hemangioendothelioma (EHE) by retrospective analysis of a national cohort of confirmed EHE cases. Authors undertook a difficult job because of – being a very rare form of cancer – the limited size of EHE samples. No wonder that they were able to identify large tumor size and high mitotic count as prognostic factors in unifocal EHE. These parameters were associated with a progressive course of unifocal disease. Lymph node metastases were mainly associated with a head-and-neck localization. Multifocal and metastatic EHE were associated with dismal clinical outcome. Authors findings confirmed the results of previous studies of Shibayama et al, Deyrup et al., and Rosenbaum et al. It might have been interesting to compare the prognostic value of the two different driver mutations resulting in EHE development, i.e. WWTR1-CAMTA1 fusion-positive and YAP-TFE3-positive lesions. Although TAZ and YAP are two paralogous transcriptional activators, the C terminal fusion partners, CAMTA1 and TFE3 are dissimilar (Merritt N et al. eLife 2021; 10:e62857.). Unfortunately, only two of the investigated 57 cases were TFE3-positive not allowing a statistically meaningful comparison of WWTR1-CAMTA1 vs. YAP-TFE3. Some patients with multifocal EHE localization and with lymph node metastases underwent „systemic therapy” (lines 173, 174, and 176). Authors should disclose the exact composition of „systemic therapy” and if these patients received anti-angiogenic drugs.

Author Response

Dear reviewer, 

Respons point 1: the reviewer makes a very interesting suggestion about comparing the TFE3 and CAMTA1 subgroup. Unfortunately, as the reviewer points out our TFE3 positive population is to small to perform any analyses.  

Respons point 2: we thank the reviewer for this suggestion and agree that this was missing within our first version of the manuscript. Therefore we added a supplementary table showing the treatment specifications.